# An improved efficient anonymous authentication with conditional privacy-preserving scheme for VANETs

Eko Fajar Cahyadi[1,2], Min-Shiang Hwang[1,3]*

**1** Department of Computer Science and Information Engineering, Asia University, Taichung, Taiwan (R.O.C.), **2** Faculty of Telecommunication and Electrical Engineering, Institut Teknologi Telkom Purwokerto, Purwokerto, Indonesia, **3** Department of Medical Research, China Medical University Hospital, China Medical University, Taichung, Taiwan (R.O.C.)

* mshwang@asia.edu.tw

**Data Availability Statement:** The research is in theorem analysis rather than empirical analysis. All information necessary to replicate the findings of the study can be found in the manuscript.

## Abstract

The study of security and privacy in vehicular ad hoc networks (VANETs) has become a hot topic that is wide open to discussion. As the quintessence of this aspect, authentication schemes deployed in VANETs play a substantial role in providing secure communication among vehicles and the surrounding infrastructures. Many researchers have proposed a variety of schemes related to information verification and computation efficiency in VANETs. In 2018, Kazemi *et al.* proposed an evaluation and improvement work towards Azees *et al.*'s efficient anonymous authentication with conditional privacy-preserving (EAAP) scheme for VANETs. They claimed that the EAAP suffered from replaying attacks, impersonation attacks, modification attacks, and cannot provide unlinkability. However, we also found out if Kazemi *et al.*'s scheme suffered from the unlinkability issue that leads to a forgery attack. An adversary can link two or more messages sent by the same user by applying Euclid's algorithm and derives the user's authentication key. To remedy the issue, in this paper, we proposed an improvement by encrypting the message using a shared secret key between sender and receiver and apply a *Nonce* in the final message to guarantee the unlinkability between disseminated messages.

## Introduction

VANETs are loaded with intelligent transportation system (ITS) properties, which make all vehicles on the road could communicate with each other via vehicle-to-vehicle (V2V) communications and to infrastructure alongside the road by vehicle-to-infrastructure (V2I) communications [1–5]. It comprises three primary entities, *i.e.*, trusted authority (TA), roadside unit (RSU), and the on-board unit (OBU) (see Fig 1). TA acts as the trust and security management center of the entire VANETs entities. Its job includes registration and parameters generation for RSUs and OBUs after joining the networks [6–9]. Meanwhile, RSUs are semi-trusted fixed infrastructures located along the road at dedicated locations and fully controlled by TA [10]. They act as a bridge between TA and vehicles (OBUs). An OBU is equipped in every vehicle

**Funding:** The Ministry of Science and Technology partially supports this research, Taiwan (ROC), under contract no.: MOST 109-2221-E-468-011-MY3. The funders had no role in study design, data collection and analysis, decision to publish, or preparation of the manuscript. There was no additional external funding received for this study.

**Competing interests:** The authors have declared that no competing interests exist.

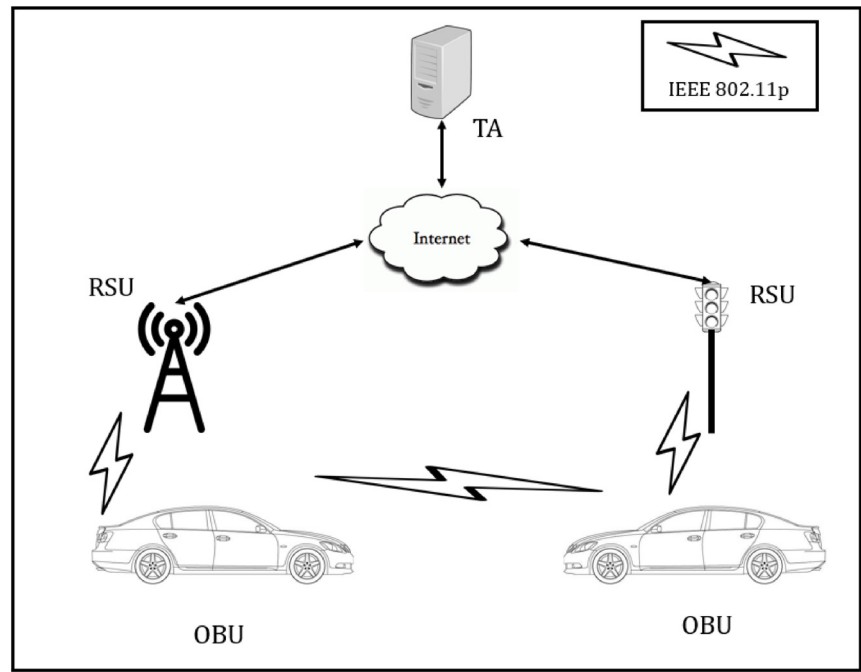

**Fig 1. The topology of VANETs.**

as a transceiver unit. It could broadcast a traffic-related message such as position, speed, and direction, to hundreds of other vehicles or RSUs every 100-300 ms [11].

One of the security aspects of VANET is against its malicious software [12–15]. The attacker accesses the vehicle's network through wireless communication and uses malicious viruses to conduct malicious attacks. These malicious viruses interfere with normal vehicle communication, deceive or tamper with information, and will seriously threaten the security of the Internet of Vehicles [12, 14].

The security and privacy aspects of the information dissemination process in VANETs strongly rely on its authentication scheme. Some popular technology, such as cloud computing, improves efficiency with its abundant storage and computing resources [16, 17]. The OBUs that storing private information are generally required to verify about 1000-5000 messages per second with about 100-500 vehicles in their communication range. At this condition, cloud-assisted VANET has greatly benefited OBUs to cope with the heavy computational tasks loading and improving road safety and traffic efficiency [18]. However, cloud providers normally use relational databases for storing metadata, which is vulnerable to being violated from users' data privacy point of view. Authors in [19] proposed a framework based on database schema redesign and dynamic reconstruction of metadata to ensure that the privacy of cloud users' data does not get compromised by an adversary even if they succeed in gaining access to the associated metadata.

Encrypting user information does not guarantee the security of user privacy, and the query itself may reveal the user's location information and identity [20, 21]. Aiming at the privacy and security issues of location-based services in the Internet of Vehicles environment. Xie *et al.* proposed a PPA-IOV privacy protection algorithm based on the analysis of LBS privacy security technology, combined with K area and pseudonym anonymity technology [21].

To enhance security, many researchers also have included biometrics, such as finger-prints and face patterns, to makes it difficult for an adversary to forge the legitimation of the user [22]. Moreover, its attributes have various desirable properties regarding personal authentication, including dependability, convenience, universality, etc. These characteristics have made biometrics adaptable to general use in authentication systems [23]. Based on the employed cryptographic mechanism, Lu *et al*. [6] distinguished the privacy-preserving authentication scheme of VANETs into five categories, including public key infrastructure (PKI)-based [11, 24–28], symmetric cryptography-based [29–31], identity (ID)-based signature [32–37], certificateless signature (CLS)-based [38, 39], and group signature-based [8, 9, 40].

In the PKI-based, the main property is each user uses a pair of cryptographic keys: a public and a private key, where the mechanism itself strongly relies on the computational complicacy for private key generation from its corresponding public key. In 2007, Raya-Hubaux [11] proposed a PKI-based CPPA scheme, which used a pair of public-private keys and corresponding certificates to hide a vehicle's real identity. However, this approach has two main drawbacks: a significant verification overhead and large storage requirements on OBU's side and a large certification revocation list (CRL) generation that making the revocation process ineffective on TA's side. In 2008, Lu *et al*. [24] proposed an efficient conditional privacy preservation (ECPP) protocol to rectify the CRL and the storage space limitations issue, where the vehicle depends on the RSU to obtain a short-term pseudonym. In the same year, Zhang *et al*. [25] also proposed an efficient RSU-aided message authentication (RAISE) protocol based on a *k*-anonymity and a hash message authentication code. In 2016, Rajput *et al*. [26] suggested a privacy-preserving hierarchical pseudonymous-based authentication protocol to resolve these PKI-based drawbacks. This protocol avoids CRL management by presenting hierarchical pseudonyms that differentiate one user from another regarding their time to live.

In 2017, Azees *et al*. [27] proposed a PKI-based efficient anonymous authentication scheme with a conditional privacy-preserving (EAAP) scheme for VANETs. Thus, the vehicles and RSUs can generate their anonymous certificates to provide privacy, and TA doesn't require to store them. Meanwhile, in case of dispute, TA can revoke a misbehaving vehicle's anonymity and disclose its real identity. As a result, the scheme was declared secure against impersonation attacks, bogus message attacks, message modification attacks, and providing privacy preservation and anonymity during the authentication of vehicles and RSUs. However, in 2018, Kazemi *et al*. [28] published an article that pointed out some weaknesses in the EAAP. First, they claimed if the scheme is vulnerable to replay attacks. The final message sent by the authorized users is not changed until the vehicle's public key is updated, so the message is not fresh and can be used several times by the adversary $\mathcal{A}$. The scheme also claimed to suffer from message modification and impersonation attacks because there are possibilities for $\mathcal{A}$ to modify the message's content or generate fake verifiable messages and generate a valid message on behalf of an authorized vehicle, respectively. Lastly, the scheme cannot provide unlinkability since the authentication key (*AK*) is not changed. This condition causing a part of the generated challenger is also fixed in different transmissions. In this way, $\mathcal{A}$ can track the sender and find out its location in the network. Therefore, Kazemi *et al*. proposed an improvement work towards the EAAP scheme.

Unfortunately, in this article, we prove if Kazemi *et al*.'s [28] scheme is also cannot provide unlinkability, which leads to forgery and impersonation attacks. By applying Euclid's algorithm, an adversary $\mathcal{A}$ can link two or more messages sent by the same user $u_i$. By deriving the user's dummy identity $DID_{u_i}$ and their partial private key $E_{u_i}$, $\mathcal{A}$ is able to generate the

authentication key $AK_{u_i}$ of $u_i$. Hence, by encrypting the message $M$ using shared secret key $E_{sk}$ between sender and receiver, and applying a *Nonce* instead of a timestamp $T$ in the final message *msg*, our improved scheme can provide unlinkability and so withstand the forgery attacks. The outline of our main contributions are as follows:

- We point out some security flaws in Kazemi *et al.*'s scheme

- We propose an improvement to address the weaknesses in Kazemi *et al.*'s scheme

- We demonstrate that our improvement is secure against forgery and impersonation attacks, and can provide unlinkability.

For a better understanding, the rest of this paper is arranged as follows. First, in Section 2, we provide preliminaries. Then, a brief review of Kazemi *et al.*'s scheme and its cryptanalysis are described in Section 3. Next, in Section 4, we propose our improvement to Kazemi *et al.*'s scheme, then analyze it in Section 5. Finally, the conclusion is conveyed in Section 6.

## Preliminaries

This section introduces the system design, adversary model, and security-privacy requirements that have to be fulfilled in VANETs.

### System model

The two-layer concept in VANETs, with TA on the top, while RSUs and OBUs on the lower layer, as seen in Fig 1, have been introduced by [32], and then used by several works [33, 34, 36] since then. The task and function of each entity have been briefly described in Section 1. For advance, the OBU in the vehicle will have communication sensors, tamper-proof device (TPD), DSRC communication medium, event data recorder (EDR), smart card and fingerprint devices, and human-machine interface to calculate an effective decision movement, as shown in Fig 2 [22]. In our VANETs ecosystem, we assume:

1. TA is uncompromised

2. Only TA that can reveal the real identity of RSUs and OBUs

3. TA—RSU communicate through a secured wireline networks

4. RSUs are semi-trusted

5. TPD is assumed to be credible.

### Adversary model

We assume that all RSUs and OBUs in the networks are not fully trusted, and the open wireless communication channel is naturally not secure. In [28], an adversary $\mathcal{A}$ is capable of performing the following actions:

1. Upon receiving two or more messages from the same user $u_i$, an $\mathcal{A}$ can relate the encrypted dummy identities $DID_{u_i}$ and encrypted OBU's private keys $E_{u_i}$ to obtain the authentication key $AK_{u_i}$ of $u_i$.

2. An $\mathcal{A}$ may forge the authentication key $AK_{u_i}$ of $u_i$ and impersonate it when entering a new TA's region.

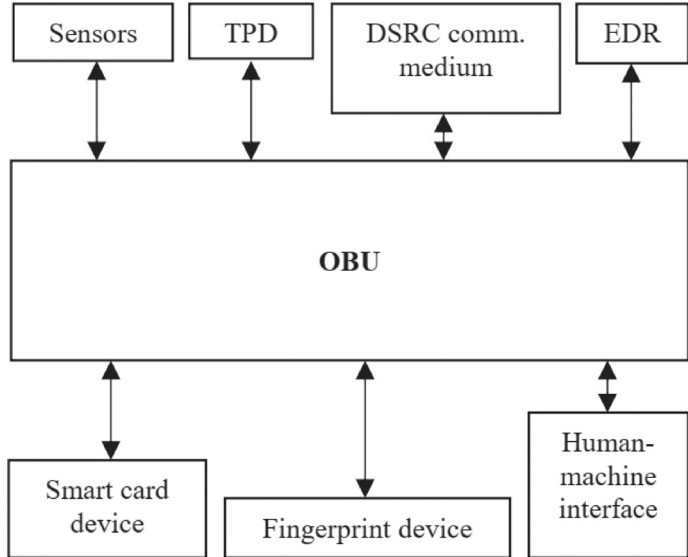

**Fig 2. Components in vehicle's side.**

## Security requirements

To design a secure authentication mechanism, first, we need to define the security and privacy requirements that must meet in VANETs.

1. Message authentication: The networks could ensure that data is sent and signed by a legitimate vehicle without being modified in transit.

2. Identity privacy-preserving: The identity of the messages' sender should be anonymous, and only TA can reveal their real identity.

3. Traceability: TA capable of revealing the real identity of the users' pseudo-identity in the case of a dispute.

4. Unlinkability: An adversary (vehicle or RSU) should not link two or more subsequent pseudonym messages of the same vehicle.

5. Resistance to impersonation attack: The networks could endure towards the attacker trying to assume or impersonate the identity of the legitimate vehicles in VANETs, to generate the signature for any messages.

6. Resistance to replaying attack: The networks could endure a passive data capture and subsequent retransmission to produce an unauthorized message by the adversaries.

## Weakness on Kazemi *et al.*'s scheme

In this section, we briefly review Kazemi *et al.*'s [28] scheme and its cryptanalysis. To comprehend the scheme's procedure, notations throughout this paper are presented in Table 1.

### Brief review of Kazemi *et al.*'s scheme

Kazemi *et al.*'s [28] scheme consists of the following six phases:

**Table 1. Notations of this paper.**

| Notation | Definition |
|---|---|
| $TA$ | Trusted authority |
| $RSU$ | Roadside unit |
| $OBU$ | On-board unit |
| $G_1$ | A group with the order of composite number $N$ |
| $p, q$ | Prime numbers |
| $g_1$ | Generator of $G_1$ |
| $h(\cdot)$ | One-way hash function |
| $a$ | Master secret keys |
| $A_1$ | Master public keys |
| $u_i$ | $i^{th}$ vehicle user |
| $OID_{u_i}$ | Original identity of $u_i$ |
| $DID_{u_i}$ | Dummy identity of $u_i$ |
| $E_{u_i}$ | Private key of OBU |
| $T$ | Timestamp |
| $sk$ | Secret key between sender and receiver |
| $Enc(\cdot)$ | Encryption algorithm |
| $M, msg$ | Original message, final message |

1. *System initialization*: The trusted authority (TA) generates a cyclic group $G_1$ with the order of a composite number $N = p \times q$, where $p$ and $q$ are large prime numbers. By considering $g_1$ is a member of $G_1$ with the order of large prime number $K$, then TA selects a hash function $h : \{0, 1\}^* \rightarrow Z_K^*$, a random numbers $a$ as its master key, and computes $A_1 = g_1^a$ as its public key. Finally, TA publishes the system parameters $\{N, g_1, G_1, A_1, h(\cdot)\}$.

2. *Registration and key generation*: Each user $u_i$ registers his/her personal information to TA. TA chooses a random number $n_i \in Z_q^*$ to generates a dummy user identity $DID_{u_i} = g_1^{n_i+a} \bmod N$, for each $u_i$. Next, TA computes the $u_i$'s private key $E_{u_i} = g_1^{-n_i} \bmod N$, and provides the authentication key $AK_{u_i} = (DID_{u_i}, E_{u_i})$ to the $u_i$. Finally, $u_i$ stores his/her $AK_{u_i}$ in the tamper-proof device (TPD), which is located in the on-board unit (OBU). The process of this phase is shown in Fig 3.

3. *Message generation*: The $u_i$ encrypts $DID_{u_i}$ and $E_{u_i}$ in $W = (DID_{u_i})^{h(M\|T)} \bmod N$ and $X = (E_{u_i})^{h(M\|T)} \bmod N$, respectively. Then, $u_i$ broadcasts the final message $msg = \langle M \| W \| X \| T \rangle$ to the other users in the network.

4. *Verify*: The receiver verifies incoming message by checking whether $W \times X = (A_1)^{h(M\|T)} \bmod N$. If the condition hold, the message would be accepted, otherwise rejected. Both of *message generation* and *verify* process are shown in Fig 4.

5. *Tracking*: In case of dispute, TA should be able to identifies $DID_{u_i}$ from its database that satisfies $W = (DID_{u_i})^{h(M\|T)} \bmod N$. In this way, the TA can find that user's identity and put it on the blacklist.

6. *Entering new TA*: When a user $u_i$ enters the new TA region, it will send its $AK_{u_i}$ to the new TA. The new TA will firstly check if $DID_{u_i} \times E_{u_i} = A_1$, to proofs if $u_i$ is already been authenticated by the previous TA. If yes, with a similar process in the *registration and key*

System initialization - Registration and key generation

| TA | User |
|---|---|
| Computes: | |
| $N = p \times q$. | |
| $A_1 = g_1^a$. | |
| $h : \{0,1\}^* \rightarrow Z_K^*$. | |
| Provides $params = \{N, g_1, G_1, A_1, h(\cdot)\}$. | |
| $\xrightarrow{\text{Sends } params}$ | |
| | Obtains $params$. |
| | Provides personal information $PI$. |
| $\xleftarrow{\text{Sends } PI}$ | |
| Computes: | |
| $DID_{u_i} = g_1^{n_i+a} \bmod q$. | |
| $E_{u_i} = g_1^{-n_i} \bmod q$. | |
| $AK_{u_i} = (DID_{u_i}, E_{u_i})$. | |
| $\xrightarrow{\text{Sends } AK_{u_i}}$ | |
| | Stores $AK_{u_i}$ in TPD. |

**Fig 3. System initialization—Registration and key generation phase in [28].**

*generation* phase, the new TA will generate a new *AK* for $u_i$ to communicate in its area. Otherwise, the new TA does not issue a new $AK_{u_i}$.

## Cryptanalysis of Kazemi *et al.*'s scheme

Assume there is one user $u_i$ broadcasts two final messages $msg = \langle M_1 \parallel W_1 \parallel X_1 \parallel T_1 \rangle, \langle M_2 \parallel W_2 \parallel X_2 \parallel T_2 \rangle$ to another users in VANETs. Referred to the Euclid's algorithm, if $a$ and $b$ are

Message generation - Verify

| User | Other users |
|---|---|
| Computes: | |
| $W = (DID_{u_i})^{h(M\parallel T)} \bmod N$. | |
| $X = (E_{u_i})^{h(M\parallel T)} \bmod N$. | |
| $msg = \langle M \parallel W \parallel X \parallel T \rangle$. | |
| $\xrightarrow{\text{Sends } msg}$ | |
| | Receives $msg$. |
| | Verify: |
| | $W \times X \stackrel{?}{=} (A_1)^{h(M\parallel T)} \bmod N$. |
| | If holds, accept $msg$, |
| | otherwise reject. |

**Fig 4. Message generation—Verify phase in [28].**

relatively prime then, there are exist integers $h'$ and $k'$ such that $h'a + k'b = 1$ [41]. By that property, if there is exit an adversary user $\mathcal{A}$ in the network receiving those set of messages sent by the same user, the encrypting process of $DID_{u_i}$ and $E_{u_i}$ in the *message generation* phase of Kazemi *et al.*'s scheme, could be impractical. The $\mathcal{A}$ can derive the secret parameter $AK_{u_i} = (DID_{u_i}, E_{u_i})$ from the set of messages in the following way:

$$W_1 = (DID_{u_i})^{h(M_1 \| T_1)} \bmod N \tag{1}$$

$$W_2 = (DID_{u_i})^{h(M_2 \| T_2)} \bmod N \tag{2}$$

$$X_1 = (E_{u_i})^{h(M_1 \| T_1)} \bmod N \tag{3}$$

$$X_2 = (E_{u_i})^{h(M_2 \| T_2)} \bmod N \tag{4}$$

In our case, if $h(M_1 \| T_1)$ and $h(M_2 \| T_2)$ are relatively prime, we can find $r$ and $s$, such that $rh(M_1 \| T_1) + sh(M_2 \| T_2) = 1$. By doing operation (5), $\mathcal{A}$ can acquire $DID_{u_i}$ based on (1) and (2).

$$\begin{aligned} W_1^r \times W_2^s &= (DID_{u_i})^{rh(M_1\|T_1)+sh(M_2\|T_2)} \bmod N \\ &= DID_{u_i} \end{aligned} \tag{5}$$

Similarly, $\mathcal{A}$ can derive $E_{u_i}$ with operation (6) based on (3) and (4):

$$\begin{aligned} X_1^r \times X_2^s &= (E_{u_i})^{rh(M_1\|T_1)+sh(M_2\|T_2)} \bmod N \\ &= E_{u_i} \end{aligned} \tag{6}$$

From (5) and (6), any adversary $\mathcal{A}$ can obtain the secret parameter $AK_{u_i} = (DID_{u_i}, E_{u_i})$ of the legal user $u_i$ by intercepting two messages $msg = \langle M_1 \| W_1 \| X_1 \| T_1 \rangle, \langle M_2 \| W_2 \| X_2 \| T_2 \rangle$. Furthermore, by this weakness, after deriving $AK_{u_i}$ from any user, the adversary $\mathcal{A}$ can impersonate them when entering a new TA's region.

## Improvement of Kazemi *et al.*'s scheme

After adversary $\mathcal{A}$ gets $(M_1, T_1)$ and $(M_2, T_2)$ by intercepting messages $msg = \langle M_1 \| W_1 \| X_1 \| T_1 \rangle, \langle M_2 \| W_2 \| X_2 \| T_2 \rangle$ from user $u_i$ to RSU or other users $u_j$, $\mathcal{A}$ calculates the hash functions $h(M_1 \| T_1)$ and $h(M_2 \| T_2)$. Next, $\mathcal{A}$ can find $r$ and $s$ through the Euclid's algorithm, so that $rh(M_1 \| T_1) + sh(M_2 \| T_2) = 1$. Finally, $\mathcal{A}$ derives the secret parameter $AK_{u_i} = (DID_{u_i}, E_{u_i})$ of the legitimate user $u_i$.

To overcome this problem, we have proposed two types of improvements. The first is to use the shared key $sk$ between the sender and the receiver to encrypt the original message $M$ in the *message generation* phase. The second alternative is to use *Nonce* instead of the timestamp $T$ in the *message generation*, *verify*, and *tracking* phases.

### First type improvement

In the first type improvement, the *system initialization*, *registration and key generation*, and *entering new TA* phases remain the same with Kazemi *et al.*'s [28]. Meanwhile, we modify the *message generation* phase, therefore the calculation in *verify*, and *tracking* phases also follow.

1. *Message generation*: The $u_i$ encrypts $DID_{u_i}$ and $E_{u_i}$ in $W = (DID_{u_i})^{h(Enc_{sk}(M)\|T)} \bmod N$ and $X = (E_{u_i})^{h(Enc_{sk}(M)\|T)} \bmod N$, respectively. Then, $u_i$ broadcasts the final message $msg = \langle Enc_{sk}(M) \| W \| X \| T \rangle$ to the other users in the network.

2. *Verify*: The receiver verifies *msg* by checking whether $W \times X = (A_1)^{h(Enc_{sk}(M)\|T)} \bmod N$. If the condition hold, the message would be accepted, otherwise rejected.

3. *Tracking*: In case of dispute, TA should be able to identifies $DID_{u_i}$ from its database that satisfies $W = (DID_{u_i})^{h(Enc_{sk}(M)\|T)} \bmod N$. In this way, the TA can find that user's identity and put it on the blacklist.

In this type of improvement, the adversary $\mathcal{A}$ can intercepts broadcasted $msg = \langle Enc_{sk}(M) \| W \| X \| T \rangle$ but cannot perform (5) and (6). This is because the messages $M_1$ and $M_2$ are already encrypted. Thus $\mathcal{A}$ is unable to find $r$ and $s$, so that $rh(Enc_{sk}(M_1)\|T_1) + sh(Enc_{sk}(M_2) \| T_2) = 1$ is not holds.

## Second type improvement

In our second type improvement, the *system initialization*, *registration and key generation*, and *entering new TA* phases also remain the same with Kazemi *et al.*'s [28]. Meanwhile, we modify the *message generation* phase, therefore the calculation in *verify*, and *tracking* phases also follow. Remember, different from the timestamp, *Nonce* is only shared between the sender and the receiver. Once the *verify* phase is passed, the *Nonce* stored in the sender and receiver will be added by 1. The second type of improvement is given as follows:

1. *Message generation*: The $u_i$ encrypts $DID_{u_i}$ and $E_{u_i}$ in $W = (DID_{u_i})^{h(M\|Nonce)} \bmod N$ and $X = (E_{u_i})^{h(M\|Nonce)} \bmod N$, respectively. Then, $u_i$ broadcasts the final message $msg = \langle M \| W \| X \| Nonce \rangle$ to the other users in the network.

2. *Verify*: The receiver verifies *msg* by checking whether $W \times X = (A_1)^{h(M\|Nonce)} \bmod N$. If the condition hold, the message would be accepted, otherwise rejected.

3. *Tracking*: In case of dispute, TA should be able to identifies $DID_{u_i}$ from its database that satisfies $W = (DID_{u_i})^{h(M\|Nonce)} \bmod N$. In this way, the TA can find that user's identity and put it on the blacklist.

Similarly, in the second type improvement, $\mathcal{A}$ can intercept the broadcasted $msg = \langle M \| W \| X \| Nonce \rangle$, but cannot perform (5) and (6) to derive $DID_{u_i}$ and $E_{u_i}$. This is because the messages' $Nonce_1$ and $Nonce_2$ are only shared by the secrets of the sender and receiver. $\mathcal{A}$ cannot find $r$ and $s$, so that $rh(M_1 \| Nonce_1) + sh(M_2 \| Nonce_2) = 1$ is not holds.

## Analysis

This section analyzes the security and performance of our improved scheme.

## Security analysis

Our proposed improvement is relatively the same as Kazemi *et al.*'s [28] scheme, except for shared secret key *sk* and *Nonce* utilization in the *message generation* phase. Therefore, some of the security requirements analyses remain the same. We compare the security analysis between Kazemi *et al.*'s [28], Horng *et al.*'s [34], Tzeng *et al.*'s [36], and our improvement scheme in Table 2.

**Table 2. Security comparisons.**

| Requirement | [28] | [34] | [36] | Ours |
|---|---|---|---|---|
| Message integrity | V | V | V | V |
| Identity privacy-preserving | V | V | V | V |
| Traceability | X | V | V | V |
| Unlinkability | X | X | V | V |
| Resistance to impersonation attack | X | V | V | V |
| Resistance to replaying attack | V | X | V | V |

**Message authentication.** In the *verify* phase of Kazemi *et al.*'s scheme, upon receiving the final message $msg = \langle Enc_{sk}(M) \parallel W \parallel X \parallel Nonce \rangle$, receiver verifies it by checks whether $W \times X = (A_1)^{h(Enc_{sk}(M)\parallel T)} \bmod N$. Since the message $M$ is encrypted with the shared secret key $sk$ between the verifier/receiver and the sender, so its only the addressed receiver who can decrypt the $M$. In addition, by our second improvement, only intended receiver who able to get the *msg* since it is shared with *Nonce*. Therefore, $\mathcal{A}$ cannot earn $AK_{u_i}$ and forge the $u_i$.

**Identity privacy-preserving.** The identity privacy-preserving is strongly related to how the adversary $\mathcal{A}$ can derives the authentication key $AK_{u_i} = \{DID_{u_i}, E_{u_i}\}$ of the user $u_i$. As the final messages in our improvement are $msg = \langle Enc_{sk}(M)\parallel W \parallel X \parallel T \rangle$ and $msg = \langle M \parallel W \parallel X \parallel Nonce \rangle$, an adversary $\mathcal{A}$ cannot perform an infiltration to steal $DID_{u_i}$ and $E_{u_i}$ from $W = (DID_{u_i})^{h(Enc_{sk}(M)\parallel T)}$ and $X = (E_{u_i})^{h(Enc_{sk}(M)\parallel T)}$, respectively, since finding modular $e^{th}$ roots is a hard problem [28]. By this condition, the vehicle user $u_i$ in this scheme can preserve their identity privacy.

**Traceability.** The TA has a capability to derive $DID_{u_i}$ of any user $u_i$ from $W = (DID_{u_i})^{h(Enc_{sk}(M)\parallel T)}$. Meanwhile, the mapping from dummy identities $DID_{u_i}$ to original identities $OID_{u_i}$ is done only in the TA. So, only TA can reveal the real identity of all entities in the network in the case of a dispute.

**Unlinkability.** As elaborated in Section 3, by encrypting message $M$ with the shared secret key $sk$ between the sender and receiver $Enc_{sk}(M)$, any adversaries $\mathcal{A}$ cannot link the two messages sent by the same user $u_i$. When $W = (DID_{u_i})^{h(Enc_{sk}(M)\parallel T)}$ and $X = (E_{u_i})^{h(Enc_{sk}(M)\parallel T)}$ are calculated using $Enc_{sk}(M)$, both $DID_{u_i}$ and $E_{u_i}$ cannot be found by $\mathcal{A}$ using Euclid's algorithm. Hence, both operation (5) and (6) are not hold. By this condition, our improved scheme reaches the unlinkability, and $AK_{u_i} = \{DID_{u_i}, E_{u_i}\}$ is safe.

**Resistance to impersonation attack.** Since the adversary $\mathcal{A}$ cannot link two messages $W_1$ and $W_2$ described in (5), or $X_1$ and $X_2$ described in (6), it is impossible for $\mathcal{A}$ to derives $AK_{u_i} = \{DID_{u_i}, E_{u_i}\}$ and impersonate the sender $u_i$ of those messages.

**Resistance to replaying attack.** The utilization of *Nonce* in $msg = \langle M \parallel W \parallel X \parallel Nonce \rangle$, gives the receiver the latest message possible and makes it impossible for $\mathcal{A}$ to replay the *msg* since the *Nonce* only shared between two particular sender-receiver pair.

## Performance analysis

This section discusses the computational complexity of the certificate and the signature verification process of our improved scheme. Let $T_h$ the time required for performing a one-way hash function, and $T_{ep1}$ is the time needed to perform exponentiation in $G_1$. In the *verify* phase of Kazemi *et al.* [28], to authenticate the user $u_i$ and its message, the receiver is checking whether $W \times X = (A_1)^{h(M\parallel T)} \bmod N$. From Table 3, we can see the complexity of performing

**Table 3. Comparison of message verification complexity.**

| Schemes | One message | $n$ messages |
|---|---|---|
| Kazemi *et al.*'s [28] | $T_{ep1} + T_{mul} + T_h$ (2.03523 ms) | $T_{ep1} + nT_{mul} + nT_h$ (2.03+0.00523n ms) |
| Horng *et al.*'s [34] | $2T_{par} + 2T_{mul} + T_H + T_h$ (2.69686 ms) | $2T_{par} + 2nT_{mul} + nT_H + nT_h$ (2.68+0.01686n ms) |
| Tzeng *et al.*'s [36] | $2T_{par} + T_{mul} + T_h$ (2.68523 ms) | $2T_{par} + T_{mul} + T_h$ (2.68+0.00523n ms) |
| Ours | $T_{ep1} + T_{mul} + T_h$ (2.03523 ms) | $T_{ep1} + nT_{mul} + nT_h$ (2.03+0.00523n ms) |

one message verification is $T_h + T_{ep1}$ since it only requires one exponentiation in $G_1$ of $A_1 = g_1^a$ and one hash operation of $h(M \parallel T)$. Meanwhile, to verify $n$ messages, it has to perform $n(T_h + T_{ep1})$, because the scheme does not have a batch verification method to verify many messages at once. In our improved scheme, the verification cost for one and $n$ messages are the same as Kazemi *et al.*'s, since the message encryption process $Enc_{sk}$ gives no cost effect on efficiency. Besides [28], we also compared the efficiency of our improved scheme with two popular schemes [34, 36], that build on ID-based signatures. Since both schemes [34, 36] employ a bilinear map, map-to-point, and point multiplication over elliptic curve operations, they have to deal with a costly pairing operation $T_{par}$, map-to-point hash operations $T_H$, and point multiplication $T_{mul}$. We adopt an experiment in [42], which observes computation overhead in Python charm cryptographic library, on Intel Core i7-4765T 2.00 GHz and 8 GB RAM machine. The following results are obtained: $T_{par}$ is 1.34 ms, $T_{mul}$ is 5.13 $\mu$s, $T_H$ is 0.0065 ms, $T_{ep1}$ is 2.03 ms, and $T_h$ is 0.0001 ms. The comparison of computational complexity between Kazemi *et al.*'s [28], Horng *et al.*'s [34], Tzeng *et al.*'s [36], and our improvement scheme is presented in Table 3. From Tables 2 and 3, the proposed scheme is significantly better than other schemes in terms of security and message verification complexity.

## Conclusion

In this paper, we proposed an improvement towards Kazemi *et al.*'s authentication scheme. Our investigation exhibited that an adversary $\mathcal{A}$ can derive the user's secret parameter $AK_{u_i}$ through Euclid's algorithm property. Thus, it does not provide traceability and unlinkability, which leads to impersonation attacks. Since introducing two types of improvements, whether by encrypting the message $M$ with a shared secret key $sk$ between sender and receiver or putting a *Nonce* in the final message *msg*, we have proposed improving this scheme and making it secure.

## Author Contributions

**Writing – original draft:** Eko Fajar Cahyadi.

**Writing – review & editing:** Min-Shiang Hwang.

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
