## [Decision Letter · Decision Letter 0]

8 Jul 2021

PONE-D-21-19203

An improved efficient anonymous authentication with conditional privacy-preserving scheme for VANETs

PLOS ONE

Dear Dr. Hwang,

Thank you for submitting your manuscript to PLOS ONE. After careful consideration, we feel that it has merit but does not fully meet PLOS ONE’s publication criteria as it currently stands. Therefore, we invite you to submit a revised version of the manuscript that addresses the points raised during the review process.

We look forward to receiving your revised manuscript.

Kind regards,

Muhammad Khurram Khan, Ph.D.

Academic Editor

PLOS ONE

 [The Ministry of Science and Technology partially supported this research, Taiwan (ROC), under contract no.: MOST 109-2221-E-468-011-MY3 and MOST 110-2622-8-468-001 -TM1.

NO - Include this sentence at the end of your statement: The funders had no role in study design, data collection and analysis, decision to publish, or preparation of the manuscript.]. 

4. We note you have included a table to which you do not refer in the text of your manuscript. Please ensure that you refer to Table 2 in your text; if accepted, production will need this reference to link the reader to the Table.

Reviewers' comments:

Reviewer's Responses to Questions

**Comments to the Author**

1. Is the manuscript technically sound, and do the data support the conclusions?

Reviewer #1: Yes

Reviewer #2: Yes

2. Has the statistical analysis been performed appropriately and rigorously? 

Reviewer #1: Yes

Reviewer #2: N/A

3. Have the authors made all data underlying the findings in their manuscript fully available?

Reviewer #1: Yes

Reviewer #2: Yes

4. Is the manuscript presented in an intelligible fashion and written in standard English?

Reviewer #1: Yes

Reviewer #2: Yes

5. Review Comments to the Author

Reviewer #1: The authors proposed an improved efficient anonymous authentication with conditional privacy-preserving scheme for VANETs. First, the authors pointed out that the Kazemi et al.’s scheme is vulnerable to some attacks. Second, they proposed an improvement to remedy these attacks. Check its correctness and analysis, it is correct and complete. The evaluation results shows that their scheme is better in terms of computation time consumption. The paper is well-written and well-organization. However, some suggestions should be addressed if the paper is accepted.

1. Several grammar mistakes of English exist in this paper, which need to be carefully checked and revised.

2. The format of your paper is wrong. Please check your format for this journal using their templet.

3. Please cite the following related important papers in this area.

a) DOI: 10.1109/IVS.2015.7225898

b) https://doi.org/10.1007/s11276-013-0543-7

Reviewer #2: In this paper authors performed cryptanalysis of a previously published paper. Authors figured Kazemi et al.’s scheme suffered from the un-linkability issue that leads to a forgery attack. An adversary can link two or more messages sent by the same user by applying Euclid’s algorithm and derives the user’s authentication key. To address this problem, authors encrypt the message using a shared secret key between sender and receiver and apply a Nonce in the final message to guarantee

the un-linkability between disseminated messages.

The work is interesting, but there are some concerns which should be addressed for the next round of review:

1. I think, there is a room to improve the English of this paper. I suggest authors to completely proof-read this paper to improve its readability.

2. I found there are two relevant schemes in VANETs authentication which have been overlooked by authors and should be discussed in the introduction and if possible should be compared with:

https://ieeexplore.ieee.org/abstract/document/6576161

https://ieeexplore.ieee.org/abstract/document/7047920

These two papers are seminal in ID-based verification of VANETs and could be discussed in the introduction section also to give a clear background of the research work.

3. The digital authentication field has revolutionized during the last 2 decades and there are so many new innovations by using cloud and biometrics authentication. I suggest authors to discuss how privacy, cloud and biometrics have impacted the authentication field and what benefits it has brought to the automotive industry for VANETs. There are some following papers which could give a good background of it and can be discussed in the revision:

https://www.sciencedirect.com/science/article/abs/pii/S1084804512001890

https://www.sciencedirect.com/science/article/abs/pii/S1084804518301309

https://www.tandfonline.com/doi/abs/10.4103/0256-4602.50703

4. Its better to improve and revise the conclusion section of this paper.

6. PLOS authors have the option to publish the peer review history of their article (what does this mean?). If published, this will include your full peer review and any attached files.

Reviewer #1: No

Reviewer #2: No

---

## [Author Response · Author response to Decision Letter 0]

9 Aug 2021

Reviewer 1: Comments to the Author

 Several grammar mistakes of English exist in this paper, which need to be carefully checked and revised.

Authors’ reply:

Thanks for the referee’s comment. We have checked and revised some grammatical errors in our manuscript. In addition, we have tried our best to proofread it.

 The format of your paper is wrong. Please check your format for this journal using their template.

Authors’ reply:

Thanks for the referee’s comment. We use the template from the PLOS ONE journal’s website for our manuscript. https://journals.plos.org/plosone/s/latex.

 Please cite the following related important papers in this area.

a) DOI: 10.1109/IVS.2015.7225898

b) https://doi.org/10.1007/s11276-013-0543-7

Authors’ reply:

Thanks for the referee’s comment. Yes, we have included both “https://doi.org/10.1109/IVS.2015.7225898” and “https://doi.org/10.1007/s11276-013-0543-7” on the Introduction section and References [23] and [25]. We categorized the first article as a symmetric cryptography-based authentication scheme. Meanwhile, the other one is included as the ID-based signature scheme.

Reviewer 2: Comments to the Author 

 I think, there is a room to improve the English of this paper. I suggest authors to completely proof-read this paper to improve its readability. 

Authors’ reply:

Thanks for the referee’s comment. We have checked and revised some grammatical errors in our manuscript. In addition, we have tried our best to proofread it. 

 I found there are two relevant schemes in VANETs authentication which have been overlooked by authors and should be discussed in the introduction and if possible should be compared with:

https://ieeexplore.ieee.org/abstract/document/6576161

https://ieeexplore.ieee.org/abstract/document/7047920

These two papers are seminal in ID-based verification of VANETs and could be discussed in the introduction section also to give a clear background of the research work. 

Authors’ reply:

Thanks for the referee’s comment. Both papers you suggested above are important, and we also added them in References [26] and [28]. We have included both suggested articles in the Introduction section and compare them in the Security Analysis and Performance Analysis sections. In the Introduction section, both articles included an ID-based signature scheme.

 The digital authentication field has revolutionized during the last 2 decades and there are so many new innovations by using cloud and biometrics authentication. I suggest authors to discuss how privacy, cloud and biometrics have impacted the authentication field and what benefits it has brought to the automotive industry for VANETs. There are some following papers which could give a good background of it and can be discussed in the revision:

https://www.sciencedirect.com/science/article/abs/pii/S1084804512001890

https://www.sciencedirect.com/science/article/abs/pii/S1084804518301309

https://www.tandfonline.com/doi/abs/10.4103/0256-4602.50703

Authors’ reply:

Thanks for the referee's comment. Yes, we have discussed the above-suggested papers in the second and third paragraphs (Page 2) of the Introduction section. We have added references in [11], [13], and [15].

 Its better to improve and revise the conclusion section of this paper.

Authors’ reply:

Thanks for the referee’s comment. We have rewritten the Conclusion section.

Previous version:

“This article has shown that Kazemi et al.’s scheme does not provide unlinkability and suffers from a forgery attack that leads to an impersonation attack. The adversary can obtain the secret parameter 〖AK〗_(u_i ) through Euclid’s algorithm by linking two or more messages msg sent by the same user u_i. To overcome these drawbacks, we have proposed our improvements to security vulnerabilities by giving the scheme an immunity to unlinkability impendency, leading to forgery and impersonation attacks.”

Revised version:

“In this paper, we proposed an improvement towards Kazemi et al.’s authentication scheme. Our investigation exhibited that an adversary A can derive the user’s secret parameter 〖AK〗_(u_i ) through Euclid’s algorithm property. Thus, it does not provide traceability and unlinkability, which leads to impersonation attacks. Since introducing two types of improvements, whether encrypts the message M with a shared secret key sk between sender and receiver or put a Nonce in the final message msg, we have proposed improving this scheme and making it secure.”

Acknowledgement

The authors would like to thank the anonymous referee for their valuable discussions and comments.

---

## [Decision Letter · Decision Letter 1]

23 Aug 2021

An improved efficient anonymous authentication with conditional privacy-preserving scheme for VANETs

PONE-D-21-19203R1

Dear Dr. Hwang,

We’re pleased to inform you that your manuscript has been judged scientifically suitable for publication and will be formally accepted for publication once it meets all outstanding technical requirements.

Kind regards,

Muhammad Khurram Khan, Ph.D.

Academic Editor

PLOS ONE

Reviewers' comments:

Reviewer's Responses to Questions

**Comments to the Author**

1. If the authors have adequately addressed your comments raised in a previous round of review and you feel that this manuscript is now acceptable for publication, you may indicate that here to bypass the “Comments to the Author” section, enter your conflict of interest statement in the “Confidential to Editor” section, and submit your "Accept" recommendation.

Reviewer #1: (No Response)

2. Is the manuscript technically sound, and do the data support the conclusions?

Reviewer #1: (No Response)

3. Has the statistical analysis been performed appropriately and rigorously? 

Reviewer #1: (No Response)

4. Have the authors made all data underlying the findings in their manuscript fully available?

Reviewer #1: (No Response)

5. Is the manuscript presented in an intelligible fashion and written in standard English?

Reviewer #1: (No Response)

6. Review Comments to the Author

Reviewer #1: (No Response)

7. PLOS authors have the option to publish the peer review history of their article (what does this mean?). If published, this will include your full peer review and any attached files.

Reviewer #1: No

---

## [Editor Report · Acceptance letter]

1 Sep 2021

PONE-D-21-19203R1 

An improved efficient anonymous authentication with conditional privacy-preserving scheme for VANETs 

Dear Dr. Hwang:

I'm pleased to inform you that your manuscript has been deemed suitable for publication in PLOS ONE. Congratulations! Your manuscript is now with our production department. 

Kind regards, 

on behalf of

Prof. Muhammad Khurram Khan 

Academic Editor

PLOS ONE